# Factors Affecting Phage–Bacteria Coevolution Dynamics

**DOI:** 10.3390/v17020235

**Published:** 2025-02-08

**Authors:** Ghadeer Jdeed, Bogdana Kravchuk, Nina V. Tikunova

**Affiliations:** Institute of Chemical Biology and Fundamental Medicine, Siberian Branch of Russian Academy of Sciences, Prospect Lavrentieva 8, Novosibirsk 630090, Russia; semali328@gmail.com

**Keywords:** coevolution, phage therapy, arms race dynamics, fluctuating selection dynamics, phage adaptation, host range expansion

## Abstract

Bacteriophages (phages) have coevolved with their bacterial hosts for billions of years. With the rise of antibiotic resistance, the significance of using phages in therapy is increasing. Investigating the dynamics of phage evolution can provide valuable insights for pre-adapting phages to more challenging clones of their hosts that may arise during treatment. Two primary models describe interactions in phage–bacteria systems: arms race dynamics and fluctuating selection dynamics. Numerous factors influence which dynamics dominate the interactions between a phage and its host. These dynamics, in turn, affect the coexistence of phages and bacteria, ultimately determining which organism will adapt more effectively to the other, and whether a stable state will be reached. In this review, we summarize key findings from research on phage–bacteria coevolution, focusing on the different concepts that can describe these interactions, the factors that may contribute to the prevalence of one model over others, and the effects of various dynamics on both phages and bacteria.

## 1. Introduction

Bacteriophages (phages) and bacteria have adapted to each other over billions of years [1]. Their relationship exemplifies host-parasite coexistence with many unique features [2]. The continuous interaction between phages and bacteria has led to the development of bacterial defense systems against phages, as well as the evolution strategies by phages to overcome these defenses, leading to nested interactions between specialists phages and bacteria [3]. This evolutionary arms race is not the only form of interaction; in many instances, the adaptations acquired can be costly in terms of fitness or virulence. Consequently, alternative types of interactions have emerged, such as a mutually beneficial scenario in which a phage infects a bacterium, integrates its genome into the bacterial genome, and provides advantageous genes or protection against other phages [2].

Additionally, a temporal factor influences the adaptation of phages and bacteria to one another. Phages and bacteria may temporarily acquire mutations before discarding them in favor of mutations that are more suitable to contemporary counterparts, or because the mutations prove disadvantageous in certain aspects. This fluctuating selection of genotypes may be more sustainable over time and could result in the emergence of a diverse array of bacterial and phage descendants due to their ongoing interactions, creating multiple modules, within which phages and bacteria interact, while losing the ability to interact with phages and bacteria that adapted in different or previous modules [3,4].

With the rise of antibiotic resistance and the promising role that phages can play in therapy, an increasing number of studies have examined the coevolution of phages and bacteria. These studies have uncovered additional factors, dynamics, and outcomes that enhance our understanding of their interactions. This review aims to summarize studies on phage–bacteria coevolution, whether conducted under controlled *in vitro* or *in vivo* settings, or focused on testing samples collected regularly from ecosystems over extended periods. More than fifty articles published over the past twenty years (Table 1) were included in this review and the factors influencing the coevolution of phages and bacteria were analyzed. Additionally, the main features of evolutionary dynamics and the limitations of the conducted studies were described. This review does not address the various mathematical models used to depict the evolutionary dynamics, as these have been comprehensively described in literature, previously [5].

## 2. Mutation Rate and Mutation Load

When describing the changes occurring in coevolving phages and bacteria, two concepts are crucial for understanding the limitations and scope of their adaptation: mutation rate and mutation load. The mutation rate measures the number of mutations that arise in a population over time. It is typically estimated by analyzing the number of substitutions in samples from different stages of a population that is evolving over time [61]. This rate varies substantially among microorganisms. Fluctuation experiments estimate them to be ∼10^−10^ mutations per nucleotide per replication for bacteria, from 10^−7^ to 8  ×  10^−7^ mutations per nucleotide per replication for some dsDNA phages [62], and 5 ×  10^−3^ mutations per genome per replication for the phage M13 [63]. Other estimates reports it as 2 ×  10^−4^ to 4.7  ×  10^−3^ nucleotide alterations per site per year due to substitutions for virulent phages [64,65], and 1.2 × 10^−4^ substitutions per site per year for temperate phages [66]. Although these high mutation rates are thought to facilitate the rapid adaptation of viruses, beneficial mutations are generally quite rare, as most mutations tend to be neutral or harmful [67,68].

The mutational load refers to the total genetic burden within a population that results from the accumulation of deleterious mutations. Throughout the course of a population’s evolution, various damaging mutations may arise, potentially reducing the average fitness of its members and leading to the elimination of some of them. This process results in selection acting against these harmful mutations, establishing a balance between the selective pressure against them and their ongoing production through mutagenesis. At equilibrium, the frequency of a dominant deleterious mutation is represented by the ratio *m/s*, where *m* is the mutation rate and *s* is the selective disadvantage of the mutation. In this equilibrium state, the mutational load can be calculated as equivalent to the mutation rate, meaning that, as mutations occur, they contribute to the overall fitness landscape of the bacterial population, thereby influencing their ability to survive phage attacks [69].

## 3. Mechanisms of Phage–Bacteria Coevolution

Genome evolution in both phages and bacteria involves the accumulation of mutations and the acquisition of genetic material through genetic recombination. Genetic recombination occurs at two distinct scales. At the micro-scale, recombination can alter several nucleotides in a single event. At the macro-scale, recombination can result in the acquisition or deletion of entire genes or their fragments, leading to variations in gene content over time.

Phages can encode various proteins that facilitate recombination, including proteins of the Red recombination system [70], recombinases, and transposases [71]. Recombination between phages primarily occurs during coinfection, a phenomenon that has been shown to be widespread in bacterial populations [72,73,74]. So, temperate phages may acquire DNA from defective prophages within the host genome through relaxed homologous recombination [75], while lytic phages can recombine with other lytic phages or with prophages or remnants of prophages present in the host genome [76]. Evidence of genetic recombination has been identified in phage genomes [77,78], and the high variability in gene content observed in natural phage populations suggests that gene gain and loss occur relatively frequently [79]. However, the rates of these processes in the evolution of phage genomes are still unknown [64].

Interacting with phages, bacteria have developed a wide range of anti-phage defense mechanisms. These mechanisms include restriction-modification (RM) systems, abortive infection systems, protein sensing systems, toxin–antitoxin (TA) systems, argonauts, CRISPR-Cas systems, surface modifications, and others [80,81]. Some of these defense strategies prevent phages from infecting bacteria while preserving it bacterial cells. For instance, the RosmerTA system encodes a toxin that causes depolarization of the bacterial membrane, hindering the phage’s ability to absorb and inject its genome. Additionally, it produces an antitoxin that counteracts the effects of the toxin, thereby maintaining the viability of the bacteria intact [82]. Another system, known as the Hachiman system, produces two proteins: one that monitors the integrity of the bacterial genome while inactivating a second protein. When the bacterial genome is damaged, during phage infection, the second protein, which possesses nuclease activity, is activated, leading to the destruction of genetic material within the bacterial cell. Consequently, while the bacterium itself may be damaged, this mechanism serves to protect neighboring bacterial cells.

One of the most common strategies employed by bacteria to counteract phage infections is the modification of the outer membrane receptors used by phages for adsorption. For instance, *E. coli* is known to develop resistance through mutations in the regulatory gene *malT*, which suppresses the expression of the host receptor, the outer membrane protein LamB [50,83,84].

Phages, in turn, have developed specialized mechanisms to overcome the bacterial defenses. Phage λ counters the decrease in LamB expression in its host by evolving mutations in the binding domain of the host-recognition protein J, which allows it to use a new receptor, OmpF. In response, *E. coli* accumulates additional mutations in OmpF or in an inner-membrane protein complex, ManXYZ, that transports λ DNA into the cytoplasm to counter the viral infection [50,83,84].

Sometimes, phages acquire mechanisms to counteract the defense systems of their bacterial hosts. For instance, *Enterobacter cloacae* phage EC151 encodes a 7-deazaguanine modification pathway to evade the RM systems of its bacterial host [85]. Additionally, phages have evolved counter-defenses against some TA systems deployed by bacteria. Some T4-like phages produce the protein TifA, which directly binds to both the endoribonuclease ToxN (the toxin of the type III TA system) and bacterial RNA. This interaction leads to the formation of a high molecular weight ribonucleoprotein complex that inhibits ToxN [86]. So, the evolution of anti-phage systems in bacteria and the corresponding counter-mechanisms in phages represent a dynamic interplay, allowing both to adapt over time as a result of their prolonged interactions.

## 4. Coevolutionary Dynamics

Bacteria-phage coevolution is a classic example of mutual adaptation between hosts and parasites. This adaptation follows a specific dynamic that can be classified into two primary categories: arms race dynamics and fluctuating selection dynamics, commonly known as Red Queen dynamics. Additionally, other dynamics, such as leapfrog dynamics, kill-the-winner dynamics, piggyback-the-winner dynamics, and community shuffling dynamics, are also described, although they have been studied less extensively.

### 4.1. Arms Race Dynamics

Arms race dynamics are driven by directional selection favoring increasingly resistant hosts and increasingly infectious parasites, resulting in the emergence and accumulation of novel mechanisms of phage infectivity or host resistance [87,88]. The directional selection of arms race dynamics means that each novel mutation that enhances phage infectivity or bacterial resistance to phages builds upon the existing background. As coexistence continues, the ranges of resistance or infectivity expand. In this case, a hierarchical structure emerges within the population, in which each genotype represents a subset of a more generalist genotype, which is one step further down the coevolutionary race [89]. Generally, the resulting adapted host population exhibits reduced diversity and lower levels of sustained genetic variability [90], unless the cost of resistance is sufficiently high for phage-susceptible bacteria to maintain a fitness advantage.

Typically, arms race dynamics represent a short-term coevolutionary pattern that diminishes over time. The primary reason for this decline is the increasing fitness costs of generalist phages and bacteria as infectivity and resistance increase [60]. This decline is also constrained by resource consumption and the restricted number of potential mutations. After several generations of coevolution, a novel mechanism may become fixed due to recurrent selective sweeps. Subsequently, fluctuating selection dynamics can emerge, allowing for the maintenance of genetic diversity, as varying selective pressures enables multiple alleles to coexist rather than remain fixed [87,88]. While fluctuating selection dynamics can lead to the formation of modules where phages and bacteria interact independently of those in other modules, these dynamics can revert to an arms race within each separate module. This reversion creates localized patterns of coevolutionary adaptation [3].

### 4.2. Fluctuating Selection Dynamics

When costly resistance mutations are no longer beneficial to bacteria, long-term coexistence between bacteria and phages can emerge through fluctuating selection dynamics, a coevolutionary process driven by reciprocal evolutionary changes [60].

Fluctuating selection dynamics involve cyclical, long-term coevolutionary interactions, that are often associated with changes in genotype abundances [4]. These dynamics are mediated by negative frequency-dependent selection, favoring hosts that resist the most commonly encountered phage genotypes and phages that infect the most prevalent host genotypes. As phages adapt to target common genotypes, rarer genotypes become more abundant, thus restarting the cycle, sustaining a dynamic equilibrium, and leading to coexistence of multiple genotypes [60,87].

In these dynamics, if each bacterial clone evolves independently against specialized phages, each interaction will have its own evolutionary trajectory, potentially forming a distinct hierarchy of genotypes as they diversify [89]. The increased host diversity supports a broader genetic polymorphism of resistance phenotypes, which, in turn, can enhance the population’s resilience when encountering new phage genotypes [89].

It is suggested that, if phages adapt more rapidly than their hosts, fluctuating selection dynamics are likely to result in phages that are better adapted to their contemporary hosts compared to past and future hosts. Conversely, if hosts adapt faster than phages, the opposite effect is observed [16].

### 4.3. Leapfrog Dynamics

Leapfrog dynamics refer to hybrid dynamics that were suggested following the observed contradiction between the genotypic and phenotypic characteristics of bacteria and phages in co-incubation experiments [50]. These dynamics suggest that parasite genotypes with increasingly broad host ranges are selected, while hosts exhibiting heightened resistance are favored, like in the arms race dynamics. However, the diversity of hosts and parasites that arise early in the coevolutionary process is preserved, facilitating the emergence of rare individuals with advantageous phenotypes from this pool to replace the dominant strains. Consequently, while the genotypic patterns indicate fluctuating selection dynamics, the selection at the phenotypic level operates in a manner similar to the arms race dynamics model [50].

### 4.4. Kill-the-Winner Dynamics

Kill-the-Winner dynamics describe an ecological mechanism where phages and bacteria can coexist and sustain ecosystems characterized by a high diversity of strains, even under limited resources and high infection pressure. The dynamics are driven by ecological suppression of dominant bacterial population by phages, rather than reciprocal evolutionary adaptation between phages and bacteria. Generalist phages that can infect multiple hosts are crucial for both the evolution of the ecosystem—by eliminating dominant bacterial strains—and for maintaining diversity by enabling slower-growing bacteria to coexist with their faster-growing counterparts. In this scenario, viral abundance is determined by the differences in substrate affinity among the coexisting bacterial species [91,92].

### 4.5. Piggyback-the-Winner Dynamics

According to these dynamics, phages may transition from a lytic to a lysogenic lifestyle in environments characterized by high microbial abundance and growth rates, to take advantage of the success of their hosts. Switching to the temperate life cycle reduces the level of phage control over bacterial abundance and facilitates the exclusion of superinfection, thereby preventing the infection of the same bacterial cells by closely related phages. Consequently, microbial diversity declines [93]. It is important to note, however, that this observation is largely based on marine and aquatic environments and may not necessarily be representative in the human gut, for example [94].

### 4.6. Community Shuffling Dynamics

Prophages can be detrimental to their bacterial hosts, as their induction leads to host lysis [95]. The induction of phages is likely sensitive to stress, and although spontaneous induction is a rare event with minimal negative consequences for lysogen fitness, various environmental factors can transform induction from a rare and stochastic occurrence into a deterministic process [96]. In the human intestines, for example, inflammation in patients with inflammatory bowel disease is likely to increase prophage induction, contributing to intestinal dysbiosis by altering the ratio of symbionts to pathobionts, enabling pathobiont niche reoccupation, and will shuffle the microbial community [95]. In the oceans, prophages have been likened to molecular time bombs, which can be activated by changes in salinity or exposure to various pollutants [96].

## 5. Factors Affecting the Outcomes of Coevolution Experiments

Analyses of the studies included in this review revealed several factors that can influence the outcomes of coevolution (Figure 1).

### 5.1. Duration of the Experiment

Short experiments on the coevolution of bacteria and phages often result in an arms race dynamic, leading to the selection of generalist phages and bacteria that are typically weaker than their ancestral strains. Coevolving bacteria and phages favor the emergence of generalist bacteria that exhibit broader resistance to infecting phages, as well as generalist phages with wider infectivity ranges. However, the cost of resistance increases throughout the coevolutionary process. This pattern has been observed in experiments involving the coevolution of *Pseudomonas fluorescens* SBW25 and the SBW25F2 phage [97].

Over longer periods of time, the cost of resistance increases and shifts the dynamics of selection from the arms race dynamics to fluctuating selection dynamics, as can be seen from the study conducted over sixty rounds within the same *P. fluorescens* SBW25-phage SBW25F2 system [17]. In another study of bacteria-phage communities in leaves of horse chestnut trees, phages were collected monthly from eight trees over a six-month period. The finding revealed that phages were most infectious to contemporary bacterial hosts or those from the recent past compared to bacteria from the past. Notably, phages collected at the end of the season were somewhat less infectious to bacterial hosts than much earlier in the season. This pattern suggests fluctuating selection dynamics rather than the arms race dynamics [21].

Interactions between lysogens and phage sensitive strains are shaped by the antagonistic coevolution between phages and bacteria. These processes may involve key physiological traits, such as the capsule, and depend on the time frame of the evolutionary process. Over short time scales, simple and costly inactivating mutations are adaptive; however, in the long term, changes drawing more favorable trade-offs between resistance to phages and cell fitness become prevalent [58].

### 5.2. Preexisting Ability of Phages to Adapt to New Hosts and Bacteria Mutational Load

The ability of phages to adapt to non-target bacterial strains is linked to the number of point mutations required for infecting new hosts [24,97]. Expanding a phage’s host range may require predisposition to adapt to the bacterial strains present, like acquiring a single point mutation. For instance, incubation of the *Pseudomonas syringae* pv. *phaseolicola* phage F6 with three non-permissive *Pseudomonas* strains, along with its host strain, resulted in the emergence of new generalist phages after 20 passages, which were capable of infecting previously non-permissive strains [22].

The ability of bacteria to adapt to phages is influenced by their mutational load and fitness. Bacteria with high mutational loads and reduced fitness are generally less capable of developing resistance to phages, as they face greater costs in evolutionary process. For example, in *P. fluorescens* SBW25, bacterial populations with higher mutational loads experienced a more rapid decline in fitness over the course of coevolution, and their average resistance to contemporary phages was lower compared to populations with lower mutational loads. Additionally, the rate of directional coevolution was lower when phages coevolved with mutagenized bacterial populations [6].

The effect of mutational load on bacteria was also demonstrated in another study of the same phage–bacteria system. This study found that mutagenized bacteria, which had accumulated mutations induced by UV radiation, were less adaptable to phages [12].

An additional related factor in the coevolution of phages and bacteria is the relative evolutionary potential of each partner. Bacteria typically have more evolutionary potential than phages due to their larger genomes, which offer a wider array of targets for mutations. The coevolutionary system is usually ruled by the evolution of bacterial resistance. Bacteria develop phage resistance through mutations that alter or eliminate phage receptors, a process that can occur more frequently than the phage’s ability to adapt by altering its receptor-binding proteins (RBPs). This disparity in evolutionary potential often leads to bacterial resistance evolution dominating the coevolutionary dynamics [44].

### 5.3. The Bacterial Receptors Targeted by the Phage

When studying the coevolution of six different phages and their host, *Pseudomonas aeruginosa* PAO1, the dynamics varied depending on the specific phages involved, some became more infectious, while others became less infectious [24,97]. The coevolutionary dynamics were associated with the type of receptor used by the phages for infection. In the experiment, some phages used bacterial pili as their receptors, whereas others targeted lipopolysaccharides (LPS). To develop resistance against phages, the bacteria that used pili could reduce the number of pili on their surface. This reduction in pili density decreased the adsorption of pili-dependent phages. However, the loss of pili imposed a fitness cost on the bacteria as it impaired their swimming ability, making this adaptation undesirable. Over evolutionary time, this trade-off can be prevented, facilitating the fluctuating selection dynamics. Consequently, the phages that targeted pili induced these fluctuating selection dynamics, while those that bound to LPS contributed to the arms race dynamics [24].

Another study of the same bacterial strain using different phages indicated similar results [38]. Coevolving the *P. aeruginosa* PAO1 strain with two specific phages, phage 14-1, which targets LPS receptors, and phage LUZ19, which binds to the pili, resulted in receptor-dependent outcomes. *P. aeruginosa* PAO1-phage 14-1 coevolutionary system exhibited characteristics of the arms race dynamics and showed local adaptation with the emergence of several mutations in the *wzy* gene associated with the LPS receptor. In turn, *P. aeruginosa* PAO1-phage LUZ19 system displayed fluctuating selection dynamics, characterized by the presence of partial deletions in the *pilF* gene, which is associated with type IV pili [38].

### 5.4. The Presence of Additional Phages Specific to the Same Bacterial Host

The presence of multiple phages in the experiment can have dual effects on phage–bacteria coevolution. On the one hand, it can act as an evolutionary stress factor for the phages, potentially hindering their ability to adapt to their hosts. On the other hand, it can also facilitate genetic exchange through homologous or non-homologous recombination, thereby enhancing phage persistence by allowing them to evade host immunity mechanisms like CRISPR-Cas’s systems.

Using several phages that target a single bacterial strain can accelerate the emergence of bacterial resistance, if the combined lytic effects of the diverse phages are synergistic [39]. For example, infecting *P. aeruginosa* with five phages simultaneously (PEV2, LUZ19, LUZ7, 14-1, and LMA2), which targeted various cell surface receptors (LPS, Ton-B-dependent receptors or type IV pili), shifted the dynamics from fluctuating selection to arms race dynamics. This shift occurred because the diversity of the phages accelerated host evolution by enhancing the selection of resistant hosts [39]. Similarly, comparisons of combinations of two to five phages against *Pseudomonas syringae* demonstrated that a higher number of diverse phages led to a more rapid emergence of bacterial resistance. Interestingly, the use of a mixture of two phages was accompanied by bacterial resistance and counter resistance from the phage. In contrast, the five-phage mixture potentially operated without arms race dynamics and fluctuations in resistance. The addition of more phages to the mix resulted in quicker emergence of phage resistance, suggesting that the phages may compete with one another for attachment sites on the bacterial cell surface, thereby reducing their overall efficacy [55].

Conversely, the presence of multiple phages that target the same host can help them evade CRISPR-Cas immunity by facilitating the formation of chimeric phages through recombination. This was observed in experiments where the use of multiple phages that coevolved with *S. thermophilus* increased phage persistence by replacing sequences targeted by CRISPR. Furthermore, employing multiple phages can prolong the duration of phage-host coevolution by diluting the host’s capacity to develop sufficient immunity against any single phage population [32].

Different phages can select for unique mutations in their host. For instance, employing two genetically distinct phages against *E. faecalis* resulted in the selection of different mutations in genes encoding cell wall macromolecules necessary for phage infection, specifically PIPEF and Epa, which are required for successful infection by VPE25 and phi47, respectively [56]. Similarly, the deployment of genotypically diverse phages that target different sites on the bacteria *P. fluorescens* led to rapid initial resistance; however, there was a decline in contemporary resistance over time, contrasting with the arms race dynamics observed when using a single phage. This may be attributed to genetic constraints or the costs associated with developing resistance to multiple phages [57].

In some cases, the use of specific phages individually resulted in the emergence of bacterial resistance without any counter-adaptation from the phages or antagonistic coevolution. However, the introduction of additional phages triggered evolutionary dynamics and facilitated the adaptation of the phages. This was observed in experiments involving *Salmonella* sp. [42], and *Vibrio harveyi* phages [13], where only the application of phage cocktails led to the adaptation of both the phages and bacteria.

### 5.5. Increased Host Diversity

Although it is sometimes possible to expand the host range of a phage by including new non-permissive bacterial strains in a coevolutionary phage–bacterium system, this approach can lead to costs related to the rate of adaptation to these new hosts. In experiments adapting the *E. coli* phage øJB01 to infect several of non-permissive strains, an increase in the number and diversity of strains resulted in a slowdown of this adaptation process [45]. When comparing the coevolution of phage øJB01 to infect two previously non-permissive strains versus three non-permissive strains, the phage adapted to the two-host system exhibited better fitness. In contrast, the phage adapted in the three-host system faced greater selective pressure to maintain its ability to infect all three hosts. Additional experiments on host range expansion with a two-host experimental system indicated that the evolution of generalist or specialist phages is influenced by the ratios of the two hosts and the strength of trade-offs in fitness associated with each host [45].

### 5.6. The Presence of Bacterial Community

The presence of other bacterial species can significantly influence both the density of a specific bacterial population and its associated phage population, as well as their interactions [98]. Empirical studies have demonstrated that community presence suppresses the density of the focal host bacteria, as observed in systems involving *P. fluorescens* SBW25F2 [16] and *P. aeruginosa* with its phage PT7 [34].

Moreover, the presence of both community and phage can have synergistic effects. For example, it has been observed that, in wastewater systems, while both community and phage presence individually reduced the density of the focal host *Klebsiella* sp., their combined effect led to the extinction of that population [35]. Conversely, in some cases, the presence of other bacterial species can mitigate the impact of phages. This was demonstrated in a study that found that, while phages suppressed the density of *P. fluorescens* in the presence of other soil bacteria, they actually increased the density of the focal host when those other species were absent [16].

Additionally, studies have documented how community context influenced the evolutionary trade-offs encountered by focal bacteria. A study on *Enterococcus faecium* demonstrated that the presence of community enabled bacteria to escape a trade-off between phage resistance and virulence by promoting resistance mechanisms that did not compromise virulence [37].

### 5.7. The Continuous Presence of Phages and/or Evolutionary Naïve Host

Experiments on the expansion of phage host ranges showed that the presence of evolutionary naïve (ancestral) hosts is crucial for phage propagation and for exploring the effects of various mutations on its adaptability [45]. In a study involving *P. fluorescens* SBW25 and its phage SBW25F2, phages coevolved with their bacterial host over 24 days, adapting more rapidly compared to those evolved with a constant bacterial genotype. (such as an evolutionary naïve host). Most of the identified mutations were linked to host infection mechanisms [15]. When comparing different scenarios over twelve days (coevolution, evolution with evolutionary naïve bacteria, or alternating between these two conditions), it was generally found that phages increased their infectivity range at the cost of their growth rate. However, in the rotating scenario, some phages emerged with a higher infectivity rate without a decline in growth rate [48].

The absence of phages that induce bacterial resistance can redirect bacterial resources in other directions, thereby restoring sensitivity to phages [89]. Various clones of *P. syringae* pv. tomato DC3000 that developed resistance to five *Pseudomonas* phages acquired distinct mutations to regain sensitivity in the absence of these phages [99]. Similar results were observed in *Prochlorococcus* strains regaining sensitivity to five T7-like phages [28]. In contrast, a study on T6-resistant *E. coli* found no restoration of phage sensitivity even after 45,000 generations [100].

### 5.8. Growth Conditions

In high-resource environments, increased nutrient availability shifted evolutionary patterns from fluctuating selection to arms race dynamics. This shift was observed in a twelve-day coevolution experiment involving *P. fluorescens* SBW25 and phage SBW25F2 [25]. Additionally, an experiment with *Salmonella enterica* and phage vB_Sen_STGO-35-1 in rich media over 21 days revealed a rapid emergence of bacterial resistance in the first day. However, a subpopulation of bacteria persisted, allowing the phage to remain in the system, albeit with a multiple-fold decrease in its titer [51]. Similar dynamics were noted in an *E. coli* PP01 system, where both bacteria and phages continuously evolved in a mutual arms race [7]. In contrast, a 21-day experiment with *P. fluorescens* SBW25 and phage SBW25F2 in nutrient-poor media with tube shaking resulted in complete phage extinction [31].

In nutrient-rich conditions, shaking the tubes containing bacteria and phages can double the coevolution rate by increasing the likelihood of encounters between more infectious phages and bacteria. This interaction results in broader resistance among bacteria and greater infectivity among phages, thereby favoring the selection of generalist strains [97]. This aligns with a shift from fluctuating selection to arms race dynamics [26].

Most of the experiments described so far were conducted *in vitro*. However, a comparison of isolated *P. aeruginosa* phages 14–1 and PNM from an *in vivo* setting (post-therapy administration) with corresponding *in vitro* experiment revealed similar evolutionary dynamics. The dynamics were characterized by a rapid evolution of resistance in bacteria and limited evidence of phage evolution. Resistant bacteria from both environments exhibited decreased virulence; *in vivo*, this was associated with lower growth rates, while *in vitro* isolates showed increased biofilm formation [49].

Additionally, modifying the growth conditions by incorporating a seed bank in the form of endospores enabled bacterial species targeted by phages to maintain phenotypic diversity that might otherwise be lost during selection [53]. Furthermore, employing spatially structured swimming plates in coevolution experiments involving *E. coli* and phage T7 promotes both phenotypic and genotypic variability among phages and bacteria. This approach resulted in longer mutational pathways and revealed mutations that are rarely observed in well-mixed environments [29]. Conversely, testing the necessity of spatial isolation for the creation of diverse nested structures by coevolving *E. coli* and Φ21 in simple adaptive experiment revealed that such structures may not be necessary. Additionally, multiscale network structures can evolve rapidly under simple ecological conditions [3].

### 5.9. Implementing Phase Variation by the Bacteria

Phase variation is a crucial mechanism that enables bacteria to adapt to phage predation by generating phenotypic diversity within their population. This process involves reversible changes in specific genomic loci, often affecting surface structures like glycan modifications, which can impede phage adsorption [101]. For example, *C. jejuni* can evade certain phages through stochastic, phase-variable ON/OFF switching of its phage receptors, a process mediated by simple sequence repeats (SSR). Research indicates that phages influence the evolution of SSR-mediated phase variation [102].

Findings suggest that growth reducing counter-selection at a single PV locus can stabilize phage maintenance. Compensatory selection between different bacterial states affects the evolutionary stability of mutation rates, indicating that extreme mutation rates (either very high or very low) are evolutionarily disadvantageous. In the absence of selective pressure, phase variation evolution rates tend to evolve below the baseline mutation levels [102,103].

Phase variation also plays a significant role in several important bacterial pathogens by altering the expression of surface molecules and thereby modifying interactions with external factors [102]. For example, members of the human gut, Bacteroidetes, produce various phase-variable CPS that dictate host tropism for phages. Under phage predation, expression of non-permissive CPS variants is selected for, enhancing bacterial survival. In cases where CPS is absent, *Bacteroides thetaiotaomicron* can evade bacteriophage predation by changing the expression of eight distinct phase-variable lipoproteins. One of these lipoproteins, when constitutively expressed, confers resistance to multiple bacteriophages [104].

### 5.10. Effect of CRISPR-Cas System

While CRISPR-Cas can help the bacteria resist bacterial infections, it can also provide survival advantages to phages by accelerating rapid evolution of phage mutants. In one experiment, the presence of CRISPR-Cas system increased the mutation rate of phages by six orders of magnitude compared to the absence of CRISPR pressure [105]. Under the arms race dynamics, phages would coevolve to escape CRISPR by preferentially selecting variants with polymorphic positions at the seed or the protospacer adjacent motifs (PAM) regions. For example, directional changes in the phage genome were observed upon long-term coincubation of *S. thermophilus* DGCC7710 and phage 2972 [32]. This effect, which has also been observed in other phages, helps them to overcome sequence-specific CRISPR immunity.

Conversely, in another experiment, phage DC-56 mutants that appeared during a coevolutionary experiment were not necessarily associated with a strategy to evade CRISPR-Cas immunity. The accumulation of mutations in the PAM and protospacer regions was not favored, due to the inability of protospacer mutations to keep up with the variability of spacers that different bacteria have in the population. Consequently, phages may rely on recombineering; the presence of multiple phages can help them evade the CRISPR-Cas immunity as seen, for example when using two phages against *S. thermophilus* [32].

The CRISPR-Cas system plays a crucial role in stabilizing the outcomes of coevolutionary experiments between phages and hosts. Long-term coexistence between viruses and hosts is facilitated by a high diversity of CRISPR alleles, as observed in the experiments involving phage DC-56 [32]. This phenomenon aligns with a model that suggests CRISPR-Cas immunity stabilizes the virus–host system when the viral mutation rate is intermediate. This stability is achieved through the concept of population-distributed immunity, where a diverse array of CRISPR loci provides robust immunity across the population. As long as no single bacterial population significantly outcompetes others, phages will continue to find susceptible hosts, reducing the selective pressure to mutate their targeted protospacers or PAM regions. For instance, in a study involving a *Gordonia* population and phage DC-56, the most abundant population had a CRISPR array with only ancestral spacers, none of which perfectly matched the phage genome. Consequently, in the long term, neither the host nor the virus is driven to extinction, contrasting with laboratory coevolution experiments where CRISPR-mediated resistance led to rapid phage extinction [30,33].

### 5.11. Carrying a Prophage or a Plasmid

Temperate phages confer a fitness advantage to their hosts during competition. Prophages can enhance host fitness in co-culture by disadvantage non-lysogens. However, this prophage-mediated competitive advantage can be offset by the evolution of resistance mechanisms in competing strains [58].

Additionally, plasmid carriage was found to limit the coevolutionary dynamics between bacteria and phages. In *P. fluorescens* populations carrying plasmids, bacteria evolved lower levels of phage resistance, while phages developed lower infectivity compared to plasmid-free populations. The presence of phages led to the emergence of high frequencies of mucoid bacterial colonies in plasmid-carrying populations. Mucoidy provides a weak, partial resistance against certain phages, such as SBW25F2, which may have reduced the selective pressure for more robust resistance mutations. Plasmid carriage constrained both the evolution of bacterial resistance and phage infectivity, with plasmid-free bacteria evolving significantly higher rates of phage resistance compared to plasmid-carrying bacteria [106].

## 6. Limitations of the Studies

Our current understanding of fluctuating selection dynamics and the arms race dynamics in phage–bacteria systems is largely based on theoretical models that assume infinite population sizes and neglect environmental interactions that can change population sizes [4]. Most studies focused on *P. aeruginosa* and *P. fluorescens*-phage systems, with some work on *E. coli* phages. However, phages infecting other bacteria, such as most of the ESKAPE group, have been less studied in these contexts.

The durations and resolutions of these studies vary significantly, with most conducted over less than 30 days in laboratory settings. Long-term studies primarily involved environmental or clinical samples collected over months or years [47]. The lack of long-term controlled laboratory experiments complicates understanding the potential emergence of rare phage phenotypes or the accumulation of mutations beyond which bacterial hosts cannot adapt to phages. Conducting such studies would improve our predictions of the protentional outcomes of using phages in the treatment of chronic infections.

Testing the established evolutionary dynamics reveals several shortcomings. For instance, the arms race dynamics assume that new mutations will have synergetic and accumulating effects, predicting that phages with a wide host range will expand faster than those with a narrow range [107] and that bacterial strains with the highest resistance will continue to develop resistance to phages better or faster than other strains. However, experimental evidence shows that mutations can be non-additive, allowing rare phage clones to emerge if new mutations interact synergistically with existing ones, like mutations of the phage *λ* J protein [108,109,110]. This non-additivity of mutations was also demonstrated experimentally in *E. coli* when studying the interactions between two mutations, *malT*^−^ and Δ777, which appeared during incubation with phage *λ* [50].

Another limitation of evolutionary dynamics is the assumption that mutations have small effects. Meanwhile, large effect mutations like the *E. coli*’s *malT*^−^ can confer nearly complete resistance to some *λ* phages, making them large-effect mutations that would allow the strains, in which they appear, to dominate compared to others and contradicting the assumption of small mutation effects [50].

Another limitation is the neglect of recombination, which can suddenly increase phage fitness and contribute to the dominance of rare clones and allow it to bypass bacterial defense means such as CRISPR-Cas [37,46,105].

Reproducibility issues in coevolution experiments may stem from abiotic selection pressures, although some experiments have shown repeatable evolution at the gene level [27,44,56].

Laboratory experiments often face challenges due to well-mixed environments and rapid stagnation with little genetic variability. Addressing these limitations is crucial for predicting the outcomes of phage–bacteria interactions and improving phage therapy applications [29].

## 7. Evolutionary Consequences of Phage–Bacteria Coevolution and Their Importance for Phage Therapy

### 7.1. Impact on the Phage Host Range

Evolved phages can acquire an extended host range to the initially non-permissive strains used in the experiment, but not to strains of the same bacterial species that were not part of the experiment. Phage SBW25F2 was coevolved with its host *P. fluorescens* SBW25 and it was shown that, although all coevolved phages had a wider host range than the ancestral phage and could differentially infect coevolved variants of *P. fluorescens* SBW25, none of them could infect any of the other 150 alternative *P. fluorescens* strains, to which the phage SBW25F2 was not adapted [23]. This result indicated fundamental genetic constraints on the adaptation of this phage. In new phage clones with an extended host range, mutations were found in the tail fiber gene; however, these mutations were insufficient for the phages to infect other *P. fluorescens* strains [23]. The importance of this result is that it challenges the suggestion that phages with a broad host range may be genetically predisposed to infect hosts they have never encountered [23].

The experimental coevolution of cyanobacteria and a specific phage resulted in phages capable of infecting an initially resistant strain, which was genetically different from the cyanobacteria the phage coevolved with. However, this resistant strain was experimentally obtained from a strain that was originally sensitive to the phage; hence, coevolution did not in fact result in an increase in the number of strains that could be infected [19].

Importantly, the coevolved phages did not expand their host range in several other studies [23]. While bacteria–phage antagonistic coevolution is still common in natural populations, it has little effect on host range shifts unless there is pre-existing genetic compatibility between the host and parasite and/or the absence of secondary defense mechanisms to counter phage replication within the host [23]. Furthermore, in the absence of initial bacteria–phage compatibility, spontaneous mutation alone is unlikely to lead to phage infection, and horizontal gene transfer may be necessary [23]. This was shown in a study where phage JB01, its host strain *E. coli* O127, and three non-permissive *E. coli* strains coevolved together, and the phage adapted to those previously non-permissive hosts [45]. However, these three *E. coli* strains were not selected randomly, they were chosen based on data indicating that the studied phage requires a single point mutation to infect these strains.

Several factors may explain why coevolved phages were unable to infect all new bacterial strains: (a) The absence of suitable phage receptors on non-permissive strains. (b) In some cases, resistant strains lacked the mechanisms necessary for producing new virions after the phage genomic material was injected (electroporation did not facilitate the production of phage particles) [24]. (c) Resistant bacteria likely possess various defense mechanisms such as (RM) systems, CRISPR-Cas, superinfection exclusion, or other anti-phage systems [111].

### 7.2. Impact on the Bacterial Suppression

Coevolving phages with bacteria led to trained phages that postpone the emergence of phage resistance in bacteria. The phage λ that coevolved with *E. coli* B strain REL606 for 30 days was then compared with the ancestral phage λ. The trained phage developed the ability to bind to two bacterial receptors instead of one, as in the ancestral phage [46]. This significantly delayed the emergence of resistant bacteria; resistant bacteria emerged after 15 or more days when incubated with the adapted phage instead of 3 days when incubated with the ancestral phage [46].

Phage therapy assays have shown promising results when using trained phages. in *P. aeruginosa* lung infection model, a phage cocktail demonstrated improved clinical outcomes and reduced bacterial loads, with the joint application of evolved and ancestral phages further delaying resistance emergence [55,59]. Additionally, coevolutionary training can help phages bypass bacterial resistance mechanisms. For instance, when bacteria evolved mutations in a membrane transporter to resist a phage cocktail, a trained generalist phage was able to overcome this resistance due to its past experiences during training [112].

However, not all phage–bacteria interactions lead to an arms race or significant suppression. In some cases, especially when using single phage culture with no antibiotics, bacteria rapidly develop irreversible resistance without an apparent evolutionary struggle, as observed with *S. Enteritidis* and phage φSan23, while the addition of antibiotic or using multiple phages extends the susceptibly of the bacteria to the phage [42].

Bacteria may not evolve mutants that are completely resistant to phages, owing to a number of restrictions, such as limited resources and fitness costs. In addition, long-term coevolution could lead to the emergence of a highly phage-resistant strain with no evolutionary cost, which is reflected in the recovery of bacterial competitiveness in the fluctuating selection dynamics stage when evolving phage OMKO1 with *P. aeruginosa* [44].

### 7.3. Impact on Restoration of Antibiotic Sensitivity

Bacteria evolved phage resistance while simultaneously regaining sensitivity to amoxicillin, ampicillin, and gentamicin, influenced by phage selection pressure and bacterial competitiveness. Moreover, the impact of phage selection pressure on the trade-off between antibiotic and phage resistance was more pronounced in the arms race dynamics stage than in the fluctuating selection dynamics stage [60].

For instance, studies involving *P. aeruginosa* and the phage OMTO1 have demonstrated that arms race dynamics led to trade-off between phage and antibiotic resistance, whereas fluctuating selection dynamics does not exhibit this trade-off as clearly. This suggests that the evolutionary pressure during arms race dynamics may force bacteria to compromise on antibiotic resistance as they adapt to phage infection [44].

### 7.4. Impact on Bacteria Diversity

Phages play a vital role in controlling bacterial populations, both by promoting heterogeneous bacterial differentiation and by adapting to changes of their susceptible hosts [113]. However, their impact on bacterial evolution is complex. On the one hand, coevolution with phages can accelerate the removal of deleterious mutations from the host populations. On the other hand, phage-imposed selection for resistant hosts can lead to population bottlenecks, reducing effective population size and increasing the accumulation of deleterious mutations [6].

The emergence of phage resistance may result in fitness costs for the bacterial hosts, although these costs may vary. Some phage-resistant isolates may not experience significant reductions in pathogenicity or virulence, while others within the resulting population may be compromised. For instance, bacteria treated with single or multiple phages can develop resistance without major fitness deficits, as observed in some mutants [55].

Phages also drive adaptive responses in bacteria, such as the evolution of the mucoid phenotype in *P. fluorescens* SBW25 population during coevolution with lytic phages. Furthermore, phages can select for hosts that engage in apparent competition, influencing ecological and evolutionary dynamics [18].

## 8. Conclusions

Many factors shape the interaction dynamics between bacteria and phages (Figure 2), and, while these factors could possibly be used to create diverse bacterial communities and adapt phages to them, further research involving a broader set of phages is required. For phage therapy, several considerations must be taken into account, including ones that are rarely studied: the average treatment period for bacterial infections, the competition and synergy between phages in the cocktail, and the coevolution of microbiota occurring *in vivo*. Investigating these factors could streamline the process of developing adapted phage clones in phage banks, thereby reducing the time needed for finding suitable phages for treatment of patients.

Understanding the coevolutionary dynamics between phages and bacteria is crucial for two main reasons. First, it aids in selecting the appropriate phage preparation and administration method, as well as predicting the outcomes of phage application. Second, it enables the preparation of phages for various bacterial clones that may emerge during therapy by simulating treatment in the laboratory and adapting phages to coevolving bacterial clones. Experiments on phage–bacteria coevolution have demonstrated that phages can expand their host range; however, this occasionally leads to decreased fitness and infectivity. The coexistence of phages and bacteria is a complex and multifactorial process that depends on specific interactions between phages and bacteria. The integration of machine learning techniques with extensive sequencing data obtained following phage–bacteria interactions has the potential to enhance the dynamics models and improve their predictive capabilities, which are essential for the successful application of phages. Several machine learning algorithms are available for characterizing phage–bacteria interactions [114,115,116] employing both alignment-free and alignment-based methods [117], including ones that focus on key genes and proteins [114]. In recent years, the use of deep learning models in evolutionary studies has expanded significantly [118]. However, there remains a gap in developing models that can effectively infer phage–bacteria coevolution dynamics. This challenge is compounded by several factors: the variability in experimental conditions, the complexity of interactions between phages and bacteria, and the scarcity of comprehensive datasets. These limitations make it more difficult to accurately model and predict the intricate dynamics of phage–bacteria coevolution.

Currently, most experiments on the coevolution of phages and bacteria focused on natural phage-natural bacterium pairs. Given the application of phages in therapy, it is necessary to conduct more detailed experimental studies on the coevolution of phages and clinically important bacteria, particularly those belonging to the ESKAPE group.

Finally, rational design of phage cocktails by including evolved phages has demonstrated the ability to overcome bacterial resistance in coevolutionary experiments. Phages collected from later stages of the experiment exhibited a significantly enhanced efficiency in targeting and eliminating wild type bacteria [55]. This approach highlights the potential for developing more effective phage therapies by leveraging the adaptive capabilities of phages in response to bacterial evolution.

## Figures and Tables

**Figure 1 viruses-17-00235-f001:**
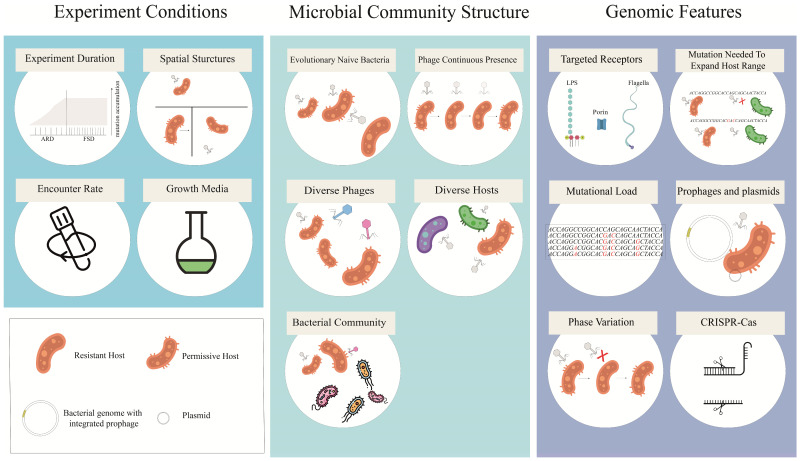
Factors influencing the coevolution of phages and bacteria. The factors can be categorized into three main groups: factors related to the presence of specific phages and bacteria, factors concerning the conditions under which coevolution occurs, and genomic factors, such as mutational load. ARD: arms race dynamics, FSD: fluctuating selection dynamics, LPS: lipopolysaccharides.

**Figure 2 viruses-17-00235-f002:**
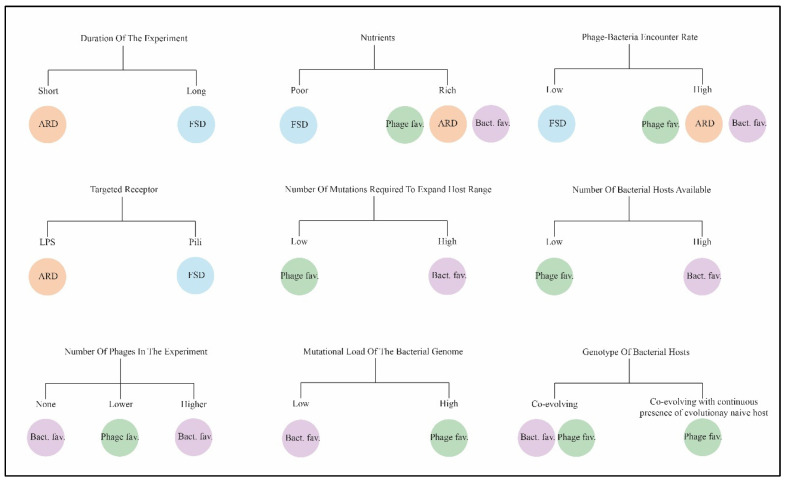
Overview of outcomes of coevolution between phages and bacteria under varying conditions. ARD—Arms race dynamics, FSD—Fluctuating selection dynamics, Bact. fav.—conditions that are favorable for bacteria adaptation; Phage fav.—conditions that are favorable for phage adaptation.

**Table 1 viruses-17-00235-t001:** Experimental studies covered in this review.

	Phages	Bacteria	Duration/ Passages	Investigated Factors	Ref.
1	SBW25F2	*Pseudomonas fluorescens*	50 transfers	Long-term selection	[6]
2	PP01	*Escherichia coli*	Continuous over 200 h	Effect of rich nutrient media in continuous culture	[7]
3	SBW25F2	*P. fluorescens*	16 transfers	Periodic shaking of microcosms	[8]
4	SBW25F2	*P. fluorescens*	16 transfers	Altering time periods between transfer to fresh microcosm	[9]
5	SBW25F2	*P. fluorescens*	8 transfers	Constant shaking of microcosms	[10]
6	SBW25F2	*P. fluorescens*	12 transfers	Mixing proportions of cultures from different microcosms prior to transfer	[11]
				Effect of simultaneous migration of coevolving bacteria and phages.	
7	SBW25F2	*P. fluorescens*	30 transfers	UV mutagenesis of ancestral bacterial clones	[12]
8	SBW25F2	*P. fluorescens*	73 transfers	Maintaining constant populations of phages across varying numbers of transfers	[13]
9	SBW25F2	*P. fluorescens*	6 transfers	Mixing proportions of cultures from different microcosms prior to transfer	[14]
				Effect of simultaneous migration of coevolving bacteria and phages	
10	SBW25F2	*P. fluorescens*	12 transfers	Effect of evolutionary naïve bacteria on the phage–bacteria coevolution	[15]
11	SBW25F2	*P. fluorescens*	66 days	Effect of the presence of other bacterial species on the evolution of a phage–bacteria system	[16]
12	SBW25F2	*P. fluorescens*	60 transfers	Long-term selection	[17]
13	SBW25F2	*P. fluorescens*	24 transfers	Effect of environment and spatial structure on the incidence of mucoid phenotype	[18]
14	RIM8	*Synchronous* spp.	6 months	Long-term selection	[19]
15	Phage lambda	*E. coli*	28 transfers	Long-term selection	[20]
16	Multiple phages unspecified	Multiple bacterial strains from various species	Several months, samples were collected once a month	Long-term selection	[21]
17	F6	Four *Pseudomonas* strains, only one permissive	20 transfers	Competition among phages for the host	[22]
18	55 clones of SBW25F2 from a previous experiment	150 *P. fluorescens* strains	Phages were used from the previous experiment	The ability of a phage that has adapted to one strain to expand its host range to additional strains that were not incubated with it	[23]
19	LMA2, PEV2, LKD16, 14-1, LUZ7, LUZ19	*Pseudomonas aeruginosa*	10 transfers	Competition among phages for the host	[24]
20	SBW25F2	*P. fluorescens*	12 transfers	Effect of rich nutrient media	[25]
21	SBW25F2	*P. fluorescens*	3 transfers over 15 days	Constant shaking of microcosms	[26]
22	T3	*E. coli*	Continuous over 30 days	Long-term selection	[27]
23	P-SSP7, P-TIP2, P-GSP1, P-SSP2, P-SSP1 P-RSP1, P-TIP1, P-TIP38, P-SSP3b, P-TIP39	3 *Prochlorococcus* strains	33 months	Long-term selection	[28]
				Effect of poor nutrient media	
24	T7	*E. coli*	2 transfers over 14 days	Spatiotemporal dynamics on swimming plates	[29]
25	DC-56	*Gordonia*	Biweekly sampling over 3 years	Long term selection in complex natural environments	[30]
26	SBW25F2	*P. fluorescens*	20 transfers	Long-term selection	[31]
27	Phage 2972	*Streptococcus thermophilus*	30 transfers	Impact of CRISPR immunization on phage genome evolution	[32]
28	Phage 2972	*S. thermophilus*	9 transfers	Impact of CRISPR immunization on phage genome evolution	[33]
29	PT7	*P. aeruginosa*	4 transfers	Effect of the presence of other bacterial species on the evolution of a phage–bacteria system	[34]
30	unnamed	*Klebsiella* sp.	Continuous over 72 h	Effect of the presence of other bacterial species and protists on the evolution of a phage–bacteria system	[35]
31	PEV2, LMA2, 14-1, LUZ7, LUZ19	*P. aeruginosa*	24 h	Competition among phages for host	[36]
32	DMS3vir and DMS3vir + acrF1	P. aeruginosa	3 transfers	Effect of the presence of other bacterial species on the evolution of a phage–bacteria system	[37]
33	14-1 and LUZ19	*P. aeruginosa*	9 transfers	Competition among phages for host	[38]
34	PEV2, LUZ19, LUZ7, 14-1, LMA2	*P. aeruginosa*	10 transfers	Competition among phages for host	[39]
35	P-SSP7, P-GSP1 and P-TIP38	5 *Prochlorococcus* strains	10 transfers over 70 days	Long term selection	[40]
36	EfV12-phi1	*Enterococcus faecium*	16 transfers	Long term selection	[41]
37	San23	*Salmonella enterica*	6 transfers	Effect of phage cocktails, individual phage and use of antibiotics	[42]
38	LP-048, LP-125	*Listeria monocytogenes*	60 h	Comparing phage cocktails and individual phages	[43]
39	OMKO1	*P. aeruginosa*	10 transfers	Effect of coevolution on restoring antibiotic sensitivity	[44]
40	FJB01	Four *E. coli* strains, one host strain and three non-permissive	10 transfers	One phage incubated with multiple potential hosts	[45]
41	Phage lambda	*E. coli*	28 transfers	Long term selection	[46]
42	ICP1, ICP2 and ICP3	*Vibrio cholerae*	32 years	Integration elements’ effects on spread of anti-phage resistance and the dynamics of evolution	[47]
43	SBW25F2	*P. fluorescens*	12 transfers	Effect of different coevolution conditions on overcoming bacterial resistance	[48]
44	14–1 and PNM	*P. aeruginosa*	6 transfers	*In vitro* versus *in vivo* evolution	[49]
45	Phage lambda	*E. coli*	37 days (from previous experiment)	Comparing phenotypic and genotypic features of trained phages	[50]
46	vB_Sen_STGO-35-1	*Salmonella Enteritidis*	21 transfers	Effect of rich nutrient media	[51]
47	20 different phages	*Erwinia, Pantoea, Pseudomonas, Stenotrophomonas*	4 years	Long-term selection from one eco-system	[52]
50	SPO1	*Bacillus subtilis*	28 days	Endospores effects on protecting the bacteria from phages	[53]
51	LQ7, ELQ7P-10, and ELQ7P-20	*Staphylococcus aureus*	20 transfers	Effect of the genetic polymorphism of minor alleles	[54]
52	MR1, MR4, MR6, MR14, MR15	*Pseudomonas syringae* pv. *syringae* (Pss)	10 transfers	Long term selection	[55]
53	VPE25 and phi47	*Enterococcus faecalis*	14 transfers	Effects of using genetically unique phages on the mutation in bacteria	[56]
54	Two SBW25F2 genotypes	*P. fluorescens*	6 transfers	Effect of phage diversity on the host resistance	[57]
55	ST14	*Klebsiella pneumoniae*	30 transfers	Long term selection	[58]
56	PWJ, WJ_Ev14	*P. aeruginosa*	5 transfers	Short term selection	[59]
57	SPO1	*B. subtilis*	14 transfers	Endospores effects on maintaining bacterial diversity	[53]
58	JNwz02	*Salmonella anatum*	30 transfers	Effect of coevolution on the trade-off between phage and antibiotic resistance in bacteria	[60]
59	Φ21	*E. coli*	24 transfers	Spatial structures effects on maintaining bacterial diversity	[3]

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
