# Peer review of "Factors Affecting Phage–Bacteria Coevolution Dynamics"

_viruses, 2025, doi:10.3390/v17020235_

Round 1
Reviewer 1 Report
Comments and Suggestions for Authors
This has the makings of an engaging paper on a topical subject. The authors have created a strong structure, but persistent problems with the quality of writing make it difficult to follow the points the paper is making. The early sections of the paper were the most difficult to parse, particularly sections that summarized key concepts in phage-bacteria co-evolutionary dynamics. Later sections that present more granular detail are easier to read.
Major comments:
1. The paper has significant and consistent errors in grammar and fluency that detrimentally impact the paper’s ability to convey the full scope of research into phage-bacteria coevolution. While some errors can be corrected through proofreading (e.g., lines 71, 72, 90: "prokaryotic argonauts" vs. "argonautes", 134, 223, 229, 308, 314-315, 323, 347, 380, 391, 404, 460), certain sections would benefit from being rewritten with the assistance of a native English speaker (e.g., lines 53, 92-95, 124, 130-132, 204-210, 233-236, 277-279, 355, 418-419, 444).
2. There needs to be more consistent engagement with references within the paper. As the paper positions itself as a comprehensive review, this should be addressed. Additionally, several key concepts are not explained well or explained using irrelevant examples from different fields. Why say that mutation rate is typically estimated using dated fossils in a paper on phage-bacterial co-evolution, and then not go on to mention any work into ancient phage? It is recommended that the authors reassess references and work to ensure that the relevance of each is made clear to the reader.
3. The section on co-evolutionary dynamics (particularly sections 4.4-4.6) is a load-bearing section of the paper. However, many sections are too brief and rely on a single citation. It is acceptable to have one citation for leapfrog dynamics, but the “kill the winner” model and community shuffling dynamics are both well-established. The paper would benefit from mentioning the origin and development of these models. With regards to “Kill the Winner”, It is recommended to mention the below publication
- Thingstad T. Frede , (2000), Elements of a theory for the mechanisms controlling abundance, diversity, and biogeochemical role of lytic bacterial viruses in aquatic systems, Limnology and Oceanography, 6, doi: 10.4319/lo.2000.45.6.1320.
4. Why aren’t there any details about the role phase variation plays in the co-existence of phage and host within an environment which enables virulent phage and host to co-exist without either being removed from an environment? Such as the human gut. For example, Bacteroides can co-exist with their phage through this mechanism. This concept has important ramifications for phage therapy concept against certain species of bacteria where these mechanisms exist
Review the following publications:
- Adaptations in gut Bacteroidales facilitate stable co-existence with their lytic bacteriophages - https://pubmed.ncbi.nlm.nih.gov/39605433/
- Phase-variable capsular polysaccharides and lipoproteins modify bacteriophage susceptibility in Bacteroides thetaiotaomicron - DOI: 10.1038/s41564-020-0746-5
Minor comments:
1. In line 69, define what an equilibrium state entails.
2. It would be informative for the audience to include more information on anti-phage defense mechanisms mentioned in lines 87-90. Anti-phage defense mechanisms are evidence of bacterial-phage co-evolution and deserve more attention in this review. Including phage counter-mechanisms for bacterial defenses in the same table would also be helpful. This table could be referred to when making the points on lines 414-416.
3. Explain acronyms in Fig. 1 and include a more informative caption.
4. Line 222, in Pseudomonas syringae pv. phaseolicola, phaseolicola should be italicized.
5. The authors should review spacing after in-text citations, which was occasionally inconsistent (e.g., line 402).
6. In lines 451-454, the discussion of machine learning reads as cursory. Multiple papers have been published on the application of machine learning techniques to phage therapy; this section of the conclusion would benefit from citing at least one of them.
Comments on the Quality of English Language
Could be improved
Reviewer 2 Report
Comments and Suggestions for Authors
This review, "Factors Affecting Phage-Bacteria Co-evolution Dynamics," addresses a critical area of study by exploring the dynamic interplay between bacteriophages and their bacterial hosts. The authors effectively emphasize the relevance of phage adaptation in combating antibiotic resistance, highlighting key models such as arms race dynamics and fluctuating selection dynamics. Furthermore, the review commendably outlines the practical implications of these interactions for phage therapy and provides an extensive summary of experimental systems, underscoring the field’s growing importance.
However, the review lacks comprehensiveness in certain areas, which detracts from its overall value. Notably, it overlooks "Rapid bacteria-phage coevolution drives the emergence of multiscale networks," a seminal work that intricately details how coevolution can shift between arms race (nested network patterns) and fluctuating selection dynamics (module networks) and toggle back again. This omission is particularly striking given the paper’s alignment with many of the ideas discussed in the review. Including this pivotal work would not only enhance the authors' understanding of the subject but also elevate the review's scholarly rigor and contextual relevance.
Additionally, the review places undue emphasis on a limited subset of study systems, particularly Pseudomonas fluorescens and its phages. While these systems are well-characterized and valuable, their dominance in the discussion restricts the broader applicability of the conclusions. Expanding the scope to incorporate underexplored bacterial hosts and ecological contexts—such as ESKAPE pathogens or environmental isolates—would significantly enrich the review. This broader perspective would provide a more nuanced understanding of the diverse factors driving coevolutionary dynamics and their implications for phage therapy.
Moreover, the writing would benefit from greater attention to precision and flow. Some sections feel repetitive or lack smooth transitions, and certain descriptions could be more succinct without sacrificing clarity. Addressing these stylistic issues would improve readability and better convey the complexity of the subject matter.
In conclusion, while this review represents a valuable contribution to the field, its impact could be amplified by addressing these critical gaps. A broader lens and the inclusion of key foundational studies are essential for providing a truly comprehensive analysis. The field of phage-bacteria coevolution is poised for transformative breakthroughs, and reviews like this play a vital role in guiding future research.
However, to fully realize this potential, it is imperative that they synthesize the full spectrum of relevant research and embrace a wider array of study systems.
Round 2
Reviewer 1 Report
Comments and Suggestions for Authors
My issues have been addressed. I have no further comments.
Author Response
Thank you for your time and effort in reviewing our manuscript. We greatly appreciate your constructive feedback and are pleased to hear that your concerns have been addressed.
Reviewer 2 Report
Comments and Suggestions for Authors
Thorough edits.
The sections on fluctuating selection and kill-the-winner dynamics are separated and do not reference each other, however, they describe the same type of dynamics. They should be linked so the reader makes this connection.
There ares still typos, this should be edited one last time.
Author Response
Comment 1: Thorough edits.
Response 1: Thank you for your time and effort in reviewing our manuscript. We greatly appreciate your constructive feedback and are pleased to hear that your concerns have been addressed.
Comment 2: The sections on fluctuating selection and kill-the-winner dynamics are separated and do not reference each other, however, they describe the same type of dynamics. They should be linked so the reader makes this connection.
Response 2: We thank you for your comment. While fluctuating selection dynamics and kill-the-winner dynamics both describe changes in population size or community composition and involve phages targeting the most abundant or faster-growing bacterial genotypes to promote the growth of slower-growing or rarer genotypes, they differ in their underlying mechanisms and temporal patterns.
Fluctuating selection dynamics are characterized by cyclical, long-term co-evolutionary processes driven by negative frequency-dependent selection, in these dynamics, both phages and bacteria co-evolve and adapt toward each other.
In contrast, kill-the-winner dynamics are not necessarily cyclical and do not rely on frequency-dependent selection. Instead, they emphasize the ecological suppression of dominant bacterial populations by phages. By targeting the most abundant or fast-growing strains, phages prevent competitive exclusion and enable the coexistence of slower-growing or less dominant strains, thereby maintaining ecosystem diversity, the phages don’t necessarily continuously adapt to bacteria in these dynamics, and they maybe generalist phages that target multiple hosts.
Thus, the sections related to fluctuations selection dynamics (4.2) and kill-the-winner dynamics (4.4) were rewritten slightly to make this difference more prominent. Thank you again for directing our attention to this issue
Comment 3: There are still typos, this should be edited one last time.
Response 3: Thank you for your comment, we did a final proof read and edit to remove the typos.